Microphytoplankton variations during coral spawning at Los Roques, Southern Caribbean

Cavada-Blanco Francoise 1 fcavada@usb.ve fcavada@gmail.com
Zubillaga Ainhoa L. 2
Bastidas Carolina 2 3
1 Laboratorio de Conservación Marino-Costera, Departamento de Estudios Ambientales, Universidad Simón Bolívar , Venezuela
2 Laboratorio de Comunidades Marinas y Ecotoxicología, Departamento de Biología de Organismos, Universidad Simón Bolivar , Caracas , Venezuela
3 Sea Grant College Program, Massachusetts Institute of Technology , United States
O’Connor Wayne
Electronic publication date: 2016 Mar 17
Publication date: 2016
Volume: 4
Electronic Location ID: e1747
Received 2015 Oct 29; Accepted 2016 Feb 9
Copyright: ©2016 Cavada-Blanco et al.
Copyright year: 2016
Copyright holder: Cavada-Blanco et al.
License: This is an open access article distributed under the terms of the Creative Commons Attribution License, which permits unrestricted use, distribution, reproduction and adaptation in any medium and for any purpose provided that it is properly attributed. For attribution, the original author(s), title, publication source (PeerJ) and either DOI or URL of the article must be cited.
License URL: https://creativecommons.org/licenses/by/4.0/

Keywords: Microphytoplankton, Coral spawning, Microheterotrophy, Los Roques, Southern Caribbean

Funding: Decanato de Investigación y Desarrollo Decanato de Estudios de Postgrado of the Universidad Simón Bolívar S1-CB-012-07 S1-IC-CB-001-09 This study was partially supported by the Decanato de Investigación y Desarrollo and the Decanato de Estudios de Postgrado of the Universidad Simón Bolívar (S1-CB-012-07 and S1-IC-CB-001-09). The funders had no role in study design, data collection and analysis, decision to publish, or preparation of the manuscript.

==============================
Phytoplankton drives primary productivity in marine pelagic systems. This is also true for the oligotrophic waters in coral reefs, where natural and anthropogenic sources of nutrients can alter pelagic trophic webs. In this study, microphytoplankton assemblages were characterized for the first time in relation to expected coral spawning dates in the Caribbean. A hierarchical experimental design was used to examine these assemblages in Los Roques archipelago, Venezuela, at various temporal and spatial scales for spawning events in both 2007 and 2008. At four reefs, superficial water samples were taken daily for 9 days after the full moon of August, including days before, during and after the expected days of coral spawning. Microphytoplankton assemblages comprised 100 microalgae taxa at up to 50 cells per mL (mean ± 8 SD) and showed temporal and spatial variations related to the coral spawning only in 2007. However, chlorophyll a concentrations increased during and after the spawning events in both years, and this was better matched with analyses of higher taxonomical groups (diatoms, cyanophytes and dinoflagellates), that also varied in relation to spawning times in 2007 and 2008, but asynchronously among reefs. Heterotrophic and mixotrophic dinoflagellates increased in abundance, correlating with a decrease of the diatom Cerataulina pelagica and an increase of the diatom Rhizosolenia imbricata. These variations occurred during and after the coral spawning event for some reefs in 2007. For the first time, a fresh-water cyanobacteria species of Anabaena was ephemerally found (only 3 days) in the archipelago, at reefs closest to human settlements. Variability among reefs in relation to spawning times indicated that reef-specific processes such as water residence time, re-mineralization rates, and benthic-pelagic coupling can be relevant to the observed patterns. These results suggest an important role of microheterotrophic grazers in re-mineralization of organic matter in coral reef waters and highlight the importance of assessing compositional changes of larger size fractions of the phytoplankton when evaluating primary productivity and nutrient fluxes.

Introduction

Phytoplankton drives the energy flow in most marine ecosystems as they are the main primary producers in environments where sufficient light allows for photosynthetic fixation of carbon (C) (Reynolds, 2006). In coral reefs, this productivity appears to be limited by nitrogen (N), as reef waters usually have minimal macronutrient concentrations (Charpy-Roubaud, Charpy & Cremoux, 1990; Furnas et al., 1990; Dizon & Yap, 1999; Koop et al., 2001; Heil et al., 2004). Nutrients and organic compounds enter the typically oligotrophic waters of coral reefs constantly through different pathways, from either natural or anthropogenic sources (Wild et al., 2004; Wild, Tollrian & Huettel, 2004; Wolanski, Richmond & McCook, 2004; Mumby, Hastings & Edwards, 2007; Hoegh-Gludberg et al., 2007). For instance, various recirculation processes in reefs drive fluxes of submarine groundwater that might transfer in-land nitrogen into the reefs (Santos et al., 2010). Other natural processes, such as mucus release by corals as well as massive or multi-specific coral spawning events, also represent influxes of labile organic compounds, from which the composition and amount of organic matter released into the water column varies greatly in space and time (Coffroth, 1990; Coffroth, 1991; Wild, Tollrian & Huettel, 2004). Anthropogenic sources of nutrients are often derived from land-based pollution, such as coastal development and agriculture on watersheds, among other activities (Burke et al., 2011).

Regardless of the source, a surplus of organic or inorganic nutrients in coral reefs can alter the ecosystem biogeochemistry and both the pelagic and benthic food webs (Koop et al., 2001; Eyre, Glud & Patten, 2008). For example, dissolved inorganic nutrients seem to be rapidly incorporated by the phytoplankton (Fabricius et al., 2013), but chronic inputs might lead to higher oxygen demand, through phyto-, bacterio- and zooplankton overgrowth and decomposition (Lapointe & Clark, 1992). Nutrient addition is a well-documented driver of taxonomical and biomass changes in marine phytoplankton assemblages (Piehler et al., 2004; Furnas et al., 2005; Howarth & Marino, 2006; Pirela-Ochoa, Troccoli & Hernández-Ávila, 2008; Strom, 2008; Mutshinda et al., 2013). Seasonal changes in phytoplankton composition in oligotrophic ocean waters, upwelling sites and coastal waters in response to both organic and inorganic nutrient inputs have been reported around the world (Casas et al., 1997; Bode et al., 2001; Oguz, Malanotte-Rizzoli & Ducklow, 2001; Vuorio et al., 2005; Wu, Sun & Zhang, 2005; Harris, 2012).

In coral reefs, phytoplankton is dominated by small-size fractions (<2 µm) such as the cyanobacteria Synechococcus and Prochlorococcus (Furnas & Mitchell 1986; Furnas et al., 2005). This small-size fraction has been the focus of studies into the response of pelagic primary producers to coral spawning events, as this response has been mainly evaluated using biomass and pigment concentrations (Eyre, Glud & Patten, 2008; Glud, Eyre & Patten, 2008; Wild et al., 2008; Apprill & Rappé, 2011; Patten et al., 2011). However, species of larger sizes (i.e., nano- and microphytoplankton: 2–200 µm), especially diatoms, also increase their primary production with nutrient inputs in coral reefs (Furnas, 1991; Crosbie & Furnas, 2001). As a result of nutrient input and cell division, larger sizes achieve greater abundances in shallow waters than in oceanic waters (Van Duyl et al., 2002); suggesting that these microalgae might also play an important role in nutrient uptake and pelagic trophic webs in coral reefs.

For example, a nutrient input after a typhoon that occurred in a reef in French Polynesia, was followed by an increase in phytoplankton biomass and primary productivity, and by a change in taxonomic composition of the microphytoplankton (Delesalle et al., 1993). Also, Glud, Eyre & Patten (2008) reported a dinoflagellate bloom in a reef flat after a coral spawning in the Great Barrier Reef; and Horne (2012) detected an increased abundance of the dinoflagellate Ceratium spp. 2 days after a spawning event in the Gulf of Mexico. As coral spawning events constitute an input of nutrients in coral reefs (Wild, Tollrian & Huettel, 2004; Eyre, Glud & Patten, 2008; Glud, Eyre & Patten, 2008; Wild et al., 2008; Patten et al., 2011), and growth of various microphytoplankton groups is nitrogen limited (Hauss, 2012), it would be expected that this input of nutrients affects the abundance and composition of larger size fraction.

Coral spawning events have been used as large natural experiments to understand the effects of an episodic organic matter input on phytoplankton primary productivity and biomass in coral reef systems (Wild et al., 2004; Wild, Tollrian & Huettel, 2004, Wild et al., 2008; Eyre, Glud & Patten, 2008; Glud, Eyre & Patten, 2008; Patten et al., 2011; Apprill & Rappé, 2011). After coral spawning, Wild et al. (2004) and Wild, Tollrian & Huettel (2004) reported an increase in sediment oxygen demand and Glud, Eyre & Patten (2008) found an increased pelagic and benthic primary production. Similarly, Eyre, Glud & Patten (2008) showed a post-spawning peak in phytoplankton biomass coinciding with the removal of dissolved inorganic nutrients and changes in nitrogen cycling on the top layer of the sediment. Nutrient additions through mass coral spawning events are thus expected to drive changes in the nitrogen and phosphorous concentrations, and consequently have effects on autotrophic and heterotrophic communities on reefs.

Reproductive timing of most common coral species in the Caribbean is well known (Szmant, 1986; Bak & Engel, 1979; Fadlallah, 1983; De Graaf, Geertjes & Videler, 1999; Budd, 1990; Bassim, Sammarco & Snell, 2002; Carlon, 2002; Brooke & Young, 2003; Beaver et al., 2004; Bastidas et al., 2005; Severance & Karl, 2006); however, the effects of this event on the reef’s pelagic trophic web have not yet been documented in the region. To contribute to this knowledge, herein we describe changes on the composition and abundance of the microphytoplankton assemblages at various spatial and temporal scales at Los Roques, Venezuela during 2007 and 2008 coral spawning events. As Los Roques is an oceanic archipelago with a marginal and local source of anthropogenic nutrients, expected coral spawning dates are a good opportunity to assess the effects of a natural source of organic nutrients on the composition and abundance of microphytoplankton separately from those of human sources.

Materials and Methods

Study site

Los Roques National Park is the most important reef complex in Venezuela, as well as in the Southern Caribbean (Casanova, Zoppy de Roa & Montiel, 2007). Los Roques is located 160 km north of the coast (66.55–66.95 W 11.65–11.98 N; Fig. 1), and the archipelago encompasses more than 50 coralline cays that are protected from wave exposure by two barrier reefs: the eastern barrier and the southern barrier, which are 20 km and 30 km long, respectively (Baamonde, 1978). Los Roques was the first marine area in Venezuela to be protected under the category of National Park and is part of the southern corridor, one of the most important areas in the Caribbean region in terms of biodiversity and extension (Rodríguez-Ramírez et al., 2008). The human population at Los Roques (approximately 2,000) is concentrated in the Northeast, specifically on Gran Roque. In the Southwest there is no permanent settlement and it is relatively less affected by recreational and fishing activities; thus, anthropogenic nutrient sources in the archipelago are marginal when compared with coastal reefs. Studies on biological and physicochemical oceanography are scarce and limited at Los Roques, however there have been studies describing temporal variability in zooplankton and phytoplankton composition and abundance at one or two reefs (González, 1989; Spiniello, 1996; Madera & Furderer, 1997), and a seasonal variation has been found in relation to the drought (November–July) and rain (May–October) regimes.

Figure 1 Map of Los Roques National Park, Venezuela, Southern Caribbean, showing the four reefs sampled at the Northeast (NE) and Southwest (SW) sectors.

Sample collection

To assess variations in microphytoplankton assemblages in relation to coral spawning at Los Roques, a hierarchical design was employed considering the following factors: (1) Locality (fixed, crossed with two levels: Northeast (NE) and Southwest (SW); (2) Reef (random, nested in locality with two levels: Gran Roque (GR) and Madrizquí (MD) on the NE and Dos Mosquises (DMS) and Cayo de Agua (CYA) on the SW); and (3) Period (fixed, crossed with three levels: before, during and after the expected spawning dates). The samples were taken from a motor boat by gently submerging a sterile 1,000 mL plastic bottle in the water down to ∼0.5–1 m depth until filled. Samples were taken by two different teams at approximately the same hour (9:00–10:00 a.m.) in the NE and SW reefs, while samples were collected less than one hour apart between reefs of the same locality. The samples were then fixed with 10% formalin, stabilized with sodium tetraborate, and closed with a sealed cap. Samples were transported to the laboratory and stored in a cold dark place until analysis. Four replicate samples were taken daily for 9 consecutive days after the full moon of August 2007 and 2008 on the basis of the expected dates of spawning. Expected dates for the first coral species that start spawning in these multi-specific events (i.e., 2–4 days after the full moon of August and September, Szmant, 1986) marked the end of the “Before” and the start of the “During” sampling period. These dates are specified for each period and year in Table 1 and were chosen based on extensive observations of coral spawning dates in the Caribbean (e.g., Van Veghel & Kahmann, 1994; Szmant et al., 1997; Sánchez et al., 1999; Mendes & Woodley, 2002) and in our study site (Bastidas et al., 2005).

Table 1 Sampling dates for the Before, During and After periods of coral spawning at Los Roques, according to the expected dates for the first coral species (Acropora palmata and A. cervicornis) that start spawning in these multi-specific events.

2007	2008	
Period	Date	Period	Date	
Before	August 28th–30th	Before	August 16th–18th	
During	August 31st–September 2nd	During	August 19th–21st	
After	September 3rd–5th	After	August 22nd–24th	

The presence of Acropora palmata larvae in the water during the sampling period of this study was further corroborated through specific antibody signals (Zubillaga, 2010). Briefly, the procedure consisted of three steps: (1) inoculation of an A. palmata larvae into rabbits; these larvae were harvested in the laboratory from bundles collected from A. palmata colonies in the field during a spawning event in 2006; (2) extraction of antibodies from rabbit’s blood samples and (3) immunological assays based on ELISA (Enzyme Linked Inmunoabsorbent Assay) to test for specificity and accuracy of the antibodies (Brian Dixon, University of Waterloo; Zubillaga, 2010). The presence of larvae in water samples was detected using a spectrophotometer that measured the colorimetric product of the enzymatic reaction between the antibodies and the coral larvae (Zubillaga, 2010). This method was preferred over direct counts due to the superior overall accuracy it has for identifying coral larval species (Carlon & Olson, 1993; Coffroth & Mulawka, 1995; Garland & Zimmer, 2002).

Environmental variables

Surface Sea Water Temperature (SST) and chlorophyll a (Chla) concentration (mg/m3) were obtained for the sampling dates by satellite image analysis (MODIS SCAR; Klein & Castillo, 2010). For this, centroids of the image cells (1 Km × 1 Km spatial resolution) that contained the sampling sites were used to download historical data. No neighbour pixels were used to obtain values of both environmental variables. Due to the proximity between MD and GR, these sampling sites in the northeast fell within the same pixel and thus, SST and Chla had the same values for these reefs.

Sample processing and data analysis

Microalgae from the water column were identified to the lowest possible taxonomic level using a Leika D MIL inverted contrast microscope. Samples were analyzed after 48 h of sedimentation in a 100 mL settling chamber. The settlement periods were used according with the Utermöhl method for oceanic/oligotrophic samples referred to in Hasle (1978). Phase contrast was used on the same sample to enhance cell detection. Magnification power used for microalgae detection (all fields were viewed) was 200×; however, 400× and 1,000× magnifications were used for identification. The abundance of microalgae species was calculated using the same protocol, based on the volume of the sample taken. Identification to the lowest possible taxonomic level was performed using Peragallo & Peragallo (1897–1908), Cupp (1943), Saunders & Glenn (1969), Ferguson Wood (1968), Marshall & Monitoring (1986), La Barbera (1984), La Barbera (1993), Sournia (1973), Sournia (1986), Sánchez-Suárez (1992a), Sánchez-Suárez (1992b), Tomas (1997), Bérard-Therriault, Poulin & Bossé (1999), Díaz-Ramos (2000) and Krayesky et al. (2009). Microalgae species were grouped into diatoms, dinoflagellates, chlorophytes, cyanophytes and coccolithophores and, based on previous work (Jeong, 1994; Jeong & Latz, 1994; Jeong et al., 2004; Jeong et al., 2005a; Jeong et al., 2005b; Jeong et al., 2005c; Jeong et al., 2007; Jeong et al., 2008; Jeong et al., 2010; Du Yoo et al., 2009; Seong et al., 2010), dinoflagellate species were further classified according to trophic functional groups into obligate autotrophic, mixotrophic and heterotrophic species.

Shannon diversity index with natural log base were calculated for every sample to facilitate comparisons among the spatial and temporal scales examined (Magurran, 2003). Microphytoplankton community structure was analysed through the attributes richness and abundance (based on the Bray–Curtis index), and composition (based on the Jaccard index). Dissimilarity matrices for both indexes were constructed from the original biological data matrices. Non-metric multidimensional ordination (nMDS) was performed to aid the visualization of temporal patterns and spatial distributions of the samples in terms of the microphytoplankton assemblage. When appropriate, centroids were used to illustrate these patterns. Null hypotheses of no differences in the abundance and composition, as well as the diversity of microphytoplankton, were tested using permutational multivariate analysis of variance (PER-MANOVA, Anderson, 2001). When significant differences (P(perm) < 0.05) were found for certain factors or interaction terms, species contributing to at least 60% of the variability between levels of the terms were identified using the SIMPER routine (Clarke & Warwick, 2001). Univariate ANOVAs were performed on the species or taxa selected by the SIMPER routine, and a posteriori pair-wise comparisons were performed between levels of the terms with p-values under a 0.05 alpha value. These analyses were performed using PRIMER-E v6 software (Clarke & Gorley, 2006).

Results

A total of 100 taxa of microalgae were identified from the water samples; 91 were identified for 2007, 51 for 2008 and 42 were common to both years (Table S1). Taking both years into account, the most abundant group were the diatoms (Bacillariophyceae), which represented 62% of the microalgae, followed by dinoflagellates (Dinophyceae, 25.4%), and Cyanophytes (Cyanophyceae, 8%). Coccolithophores (Haptophytes) and Chlorophytes represented less than 5% of the total abundance. This two-year trend of group abundances hold true for all reefs in 2008, but in 2007, cyanophytes reached highest densities of 43 cells per mL at the NE reefs only (Fig. 2). Most species of diatoms identified (62% of all counted diatoms) were small (6–10 and 16–20 µm), oceanic taxa from the genera Nitzschia, Pseudo-nitzschia, Paralia and Thalassiosira. Most taxa of dinoflagellates (48% of the counted cells) were 16–20 µm in size (Fig. 3). The coccolithophores were represented only by the species Emiliana huxleyi.

Figure 2 Average density of microphytoplankton (cells per mL) (A), Diatoms (B), Dinoflagellates (C) Cyanophytes (D), Coccolithophores (E) and Chlorophytes (F) in four reefs (GR, MD, DMS, CYA) located at the Northeast (NE) and Southwest (SW) localities of Los Roques.

B, D, A refers to Before, During and After the coral spawning events of 2007 and 2008. Density values are displayed with different scales for each of the taxonomic groups.

Microphytoplankton abundance ranged between 0.26 ± 0.06 and 49.12 ± 8.37 cells per mL (Mean ± SD) across reefs, years and spawning times. Since the variability between years was high (94% of the variance explained, PERMANOVA table not shown), the microphytoplankton assemblages were analyzed separately for each year to evaluate the effect of the other factors examined. For both years, dissimilarity in microphytoplankton abundance and composition was observed between localities (NE or solid vs. SW or empty symbols in Fig. 4), however, most of the variation in assemblage structure occurred between reefs, particularly in 2008 (Table 2). When considering all taxa, the microphytoplankton assemblage showed little variation in structure among time periods related to coral spawning (see low variation coefficients for the interaction Reef(Locality)xPeriod, Table 2). Contrasting with this weak response from the microphytoplankton structure, Shannon diversity indexes differed significantly among the spawning times in 2007 (Pseudo-F: 2.37, p = 0.042) despite, variability between replicate samples being above 60% (Fig. 5). Taxa diversity differed between the spawning times in NE reefs. At MD, there were differences Before and During spawning times (a posteriori pairwise comparisons t = 2.16, p = 0.04) and at GR, between the During and After periods (t = 1.97, p = 0.049). In 2008, diversity was similar among spawning times but varied significantly among reefs as in 2007 (Pseudo-F: 10.18, p = 0.001). In this case, diversity was lowest at CYA, a SW site, compared with the other reefs (t = 1.56, p = 0.049, respectively; Fig. 5).

Figure 3 Relative frequency of the Diatoms (A) and Dinoflagellate (B) group size classes sampled in four reefs (GR, MD, DMS, CYA) located at the Northeast (NE) and Southwest (SW) localities of Los Roques during 9 consecutive days in August and September 2007 and 2008.

Figure 4 Nonmetric Multidimentional Ordination (nMDS) based on centroids for the microphytoplankton assemblages in four reefs (GR, MD, DMS, CYA), located in the Northeast (empty symbols) and Southwest (filled symbols) of Los Roques.

nMDS based on microphytoplankton density.

Table 2 PER-MANOVA based on the Bray–Curtis dissimilarities (no transformation) of the multivariate abundance of microalgae (100 taxa); on two reefs (random, nested) in the Northeast and two reefs in the Southwest (“Locality”, fixed) of Los Roques, during, before and after (“Period”, fixed) coral spawning events in 2007 (A) and 2008 (B).

The same results were obtained for composition (Jaccard index based PERMANOVA) but are not shown.

Source of variation	d.f.	MS	Pseudo-F	P(perm)	VC (%)	
(A)	
Locality	1	20,104	3.0056	0.1664	4.88	
Period	2	2530.7	0.98108	0.4854	0.00	
Reef (Lo)	2	6690.3	3.3611	0.0001*	3.42	
LoxPE	2	2393.5	0.9279	0.5254	0.00	
Re[Lo]xPE	4	2579.7	1.296	0.07	1.28	
Residuals	240	1990.5			90.43	
Total	251					
(B)	
Locality	1	9933.6	1.3887	0.336	2.78	
Period	2	501.9	0.79077	0.5737	0.00	
Reef(Lo)	2	7153.3	12.299	0.0001*	13.14	
LoxPE	2	501.9	0.79077	0.5724	0.00	
Re[Lo]xPE	4	634.7	1.0912	0.3085	0.32	
Residuals	276	581.62			83.76	
Total	287					
Notes.

Lo Locality

PE Period of spawning times

Re Reefs

Df Degree of freedom

MS Mean Square

VC Variation Coefficient

P Probability of obtaining a Pseudo-F value similar to the ones calculated when actually there is no difference between the levels of the factors evaluated

* P(perm) < 0.05.

Figure 5 Average Shannon Diversity index values (H′ ± SD) of microphytoplankton in four reefs (GR, MD, DMS, CYA) located at the Northeast (NE) and Southwest (SW) localities of Los Roques.

B, D, A refer to Before, During and After the coral spawning events of 2007 and 2008.

Variations in microphytoplankton with respect to coral spawning times (Period) were more evident when analyzed by higher taxonomic groups instead of all taxa. The structure of diatoms, dinoflagellates and cyanophytes differed between reefs for both years, and some of these assemblages differed among spawning times at specific reefs (Table 3). In particular, the structure of dinoflagellates (Pseudo-F = 3.93, p = 0.001) and its trophic groups (Pseudo-F = 8.12, p = 0.001) differed significantly between spawning times (periods) for some reefs in 2007 (Fig. 6). Also in 2007, the assemblage of cyanophytes differed After the coral spawning at NE reefs (Pseudo-F = 2.14, p = 0.032), when the coccoid, filamentous and the nostocal Anabaena sp. peaked in an episodic manner. On the other hand, the oscillatorial cyanophytes Lyngbya and Spyrogira were identified only at Dos Mosquises (a SW reef) on the dates of expected coral spawning. While variations in dinoflagellates and cyanophytes assemblages occurred in relation to spawning times in 2007, diatoms assemblage only differed spatially among reefs for both 2007 and 2008 (Pseudo-F = 2.048, p = 0.05 and Pseudo-F = 1.75, p = 0.003, respectively).

Table 3 Statistically significant (p < 0.05) results from the PER-MANOVA analysis, based on the Bray–Curtis dissimilarities (no transformation) of the univariate abundance of microalgae taxonomic and trophic groups; on two reefs (random, nested) in the Northeast and two reef in the Southwest (“Locality”, fixed) of Los Roques, during, before and after (“Period”, fixed) coral spawning events in 2007 and 2008.

	Source of variation	Diatoms	Dinoflagellates	DTG	Cyanophytes	
2007	Locality (Lo)					
Period (PE)		*	*		
Reef[Lo]	*				
LoxPe					
Re[Lo]xPe		*	*	*	
2008	Locality (Lo)					
Period (PE)					
Reef[Lo]	*	*			
LoxPe					
Re[Lo]xPe					
Notes.

* P(perm) < 0.05.

DTG Dinoflagellate trophic groups

Figure 6 Density of Heterotrophic (A), Mixotrophic (B) and Autotrophic (C) dinoflagellates (cells per millilitre, Mean ± SD) in four reefs (GR, MD, DMS, CYA) located at the Northeast (NE) and Southwest (SW) localities of Los Roques.

B, D, A refers to Before, During and After the coral spawning event of 2007.

Most of the dissimilarity in the assemblage of dinoflagellates among spawning times in 2007 (>70% according to the SIMPER routine) was due to variations in the presence of the heterotrophic taxa Protoperidinium sp., P. thorianum, P. minutum, P. excentricum, P. conicoides, P. pyriforme, and the mixotrophic species Neoceratium lineatum and Scrippsiella trochoidea (Table S2). For each of these taxa, the univariate PERMANOVAs showed a significant difference among spawning time periods at MD and at DMS (a NE reef and a SW reef, respectively). At the SW reefs, the density of mixotrophic species decreased towards the After period of the spawning times, while the heterotrophic species increased (Fig. 6). At the NE reefs, the abundance of heterotrophic species also increased After the spawning, whereas the mixotrophic species showed an opposite pattern to the SW reefs as it increased in abundance. This pattern within locality (NE and SW) was consistent for all reefs in 2007 (Fig. 6). Correlations between the abundance of trophic groups were negative and low (≤30%), with the exception of mixotrophic and obligate autotrophs dinoflagellates (64.3%), although this was not statistically significant. In 2008, dinoflagellates as well as its trophic groups differed only at reefs scales (i.e., GR, MD, CYA, DMS), since 95% of their abundances occurred at the NE reefs where mixotrophic species peaked (Fig. 6).

For diatoms, more than 98% of the dissimilarity between the reefs in both years (SIMPER analysis) was due to variation in the presence and abundance of Aptinoptychus sp., Aulacoseria, Cerataulina pelagica, Thalassiosira subtilis, Rhizosolenia imbricata and Melosira varians. In 2007, significant differences on the abundance of C. pelagica, T. subtilis and R. imbricata occurred between reefs and between periods for R. imbricata and C. pelagica on SW reefs. Cerataulina pelagica was more abundant in the SW reefs, whereas T. subtilis was only identified on NE reefs. Significant correlations between the abundance of these diatoms and dinoflagellates trophic groups were found for SW reefs. Here, in 2007, C. pelagica abundance was negatively and weakly correlated (r2 = − 0.24; p = 0.055) with that of the mixotrophic dinoflagellate Scripsiella trochoidea, and positively correlated with that of heterotrophic dinoflagellates of the genus Protoperidinium (r2 = 0.77; p = 0.0002; Fig. 7). On the contrary, R. imbricata abundance decreased and was negatively correlated (r2 = − 0.53; p = 0.00056) with the abundance of the heterotrophic Protoperidinium species (Fig. 7). In NE reefs, none of the abundances of Thalassiosira subtilis, Cerataulina pelagica or Rhizosolenia imbricata were correlated to that of dinoflagellates trophic groups.

Figure 7 Correlation plots on abundances of Cerataulina pelagica (A) and Rhizosolenia imbricata (B) and the abundance of Protoperidinium species; corresponding to the spawning event of 2007 in two reefs (DMS, CYA) located at the Southwest (SW) sector of Los Roques.

Only statistically significant (p < 0.05) correlations, with Spearman r2 values are shown. Abundance values in cell per millilitres are displayed with different scales for each of the correlation plots.

Chlorophyll a concentrations ranged between 0.35 and 1.03 mg/m3 in 2007 and between 0.039 and 1.3 mg/m3 in 2008. These concentrations increased in days corresponding to the During and After spawning times, consistently for both years and for all reefs when data was available (Table 4A). Also, chlorophyll concentrations were more variable in 2008 than in 2007, and in the SW than in NE reefs for both years (Fig. 8). Sea surface temperatures ranged between 26.45 and 29.04°C in 2007 and between 27.4 and 29.1°C in 2008 (Table 4B). In 2007, SST increased through the sampling period in all reefs but showing a larger variability in the SW reefs (Fig. 9). In contrast, SST values were very similar among reefs in 2008, reaching higher temperatures During spawning time and decreasing by day 9 within the After spawning time (Fig. 9).

Table 4 (A) Chlorophyll a concentration (mg/m3) and (B) Seawater Surface Temperature (°C) for 9 consecutive days, “Before”, “During” and “After” the expected spawning dates for southern Caribbean acroporids.

Data obtained from remote sensing (MODIS SCAR; 1 km × 1 km resolution).

	2007	2008	
Periods	Days	NE (MD-GR)	CYA	DMS	Days	NE (MD-GR)	CYA	DMS	
(A)	
Before	28-Aug	NA	NA	NA	16-Aug	0.61	0.039	NA	
29-Aug	0.631	0.387	0.903	17-Aug	0.602	0.042	0.075	
30-Aug	0.67	0.35	NA	18-Aug	NA	0.0545	0.0544	
During	31-Aug	0.73	0.42	NA	19-Aug	0.943	0.19	0.23	
01-Sep	NA	NA	NA	20-Aug	0.903	0.280	1.076	
02-Sep	1.03	0.58	NA	21-Aug	0.87	NA	NA	
After	03-Sep	0.850	0.643	1.040	22-Aug	0.77	0.60	1.08	
04-Sep	NA	NA	NA	23-Aug	1.23	0.43	1.24	
05-Sep	0.651	0.431	0.826	24-Aug	1.30	NA	NA	
(B)	
Before	28-Aug	27.57	27.57	27.07	16-Aug	NA	NA	NA	
29-Aug	27.87	27.87	28.23	17-Aug	27.60	27.60	27.98	
30-Aug	27.72	27.72	28.09	18-Aug	27.41	27.41	27.54	
During	31-Aug	27.65	27.65	26.45	19-Aug	28.91	28.91	29.15	
01-Sep	27.82	27.82	28.92	20-Aug	28.27	28.27	28.22	
02-Sep	27.94	27.94	29.04	21-Aug	29.01	29.01	28.67	
After	03-Sep	28.40	28.40	29.03	22-Aug	28.13	28.13	27.96	
04-Sep	28.65	28.65	28.85	23-Aug	27.95	27.95	28.01	
05-Sep	28.90	28.90	28.93	24-Aug	NA	NA	NA	

Discussion

Figure 8 Surface seawater temperature (SST in °C) values obtained through remote sensing (MODIS SCAR, 1 km × 1 km resolution) for the nine-days sampling period corresponding to Before (black), During (red) and After (green) expected coral spawning times for the study site, at the Northeast (A–B) and Southwest (SW) Cayo de Agua (C–D) and Dos Mosquises (E–F) reefs of Los Roques in 2007 and 2008.

Figure 9 Chlorophyll a (Chla in mg/m3) values obtained through remote sensing (MODIS SCAR, 1 km × 1 km resolution) for the nine-days sampling period corresponding to Before (black), During (red) and After (green) expected coral spawning times for the study site, at the Northeast (A–B) and Southwest (SW) Cayo de Agua (C–D) and Dos Mosquises (E–F) reefs of Los Roques in 2007 and 2008.

Expected variations in microphytoplankton abundance and composition linked to the coral spawning events were mostly overridden by variations between years and spatial variations among reefs in Los Roques archipelago. However, when groups of the microphytoplankton were examined, dinoflagellate assemblages and trophic functional groups varied in relation to the spawning times. Heterotrophic dinoflagellates increased in abundance after the spawning event in 2007 for some reefs, while in 2008 this increment only occurred for mixotrophic dinoflagellates in one reef. Also, diatoms Cerataulina pelagica and Rhizosolenia imbricata varied simultaneously in abundance, suggesting that they may be responding to re-mineralization of dissolved nitrogen and grazing activity, respectively. These results suggest an important role of microheterotrophic grazers in re-mineralization of organic matter in coral reef waters. Also, our findings support the relevance of assessing compositional changes of larger size fractions of the phytoplankton when evaluating primary productivity and nutrient fluxes. A lack of direct measurements from the nutrient flux-cycling and benthic-pelagic coupling in our study, hampers an in-depth discussion of the patterns observed; however, changes that occurred in a few days on the composition and abundance of some taxa of the microphytoplanktonic groups examined were likely related to macronutrient inputs that resulted from coral spawning, as supported by concurrent increases in clorophyll a concentrations obtained by satellite images.

Contrasting results among studies support that phyto- and bacterioplankton and nutrient flux responses to coral spawning are variable across coral reef regions. The subtle changes in microphytoplankton at the lowest taxonomic level observed at Los Roques in relation to coral spawning, contrasted with studies from other regions (Wild et al., 2004; Wild, Tollrian & Huettel, 2004; Eyre, Glud & Patten, 2008; Glud, Eyre & Patten, 2008; Horne, 2012). In the Great Barrier Reef (GBR), Australia and the Gulf of Mexico, both planktonic and benthic microalgae blooms had been observed within 2–5 days following a mass coral spawning event (Wild et al., 2004; Wild, Tollrian & Huettel, 2004; Eyre, Glud & Patten, 2008; Glud, Eyre & Patten, 2008; Horne, 2012). At Los Roques, chlorophyll a concentrations increased During and After the spawning times in all reefs, however, only the abundance and composition of dinoflagellates and cyanophyte assemblages varied with spawning times. Furthermore, changes in cyanophytes consisted of a freshwater taxon that spiked in abundance in 2007, likely due to anthropogenic influence in the NE reefs rather than to a response from spawn material. Also, increased chlorophyll a concentrations in relation to spawning in Los Roques was lower than that reported for the GBR (Glud, Eyre & Patten, 2008). Differences in the magnitude (e.g., spawned material, number of species and geographic scale) of the spawning events influence the amount of organic matter released to the water column, and thus affect regional differences, which support the uniqueness of the mass coral spawning event at the GBR compared to that occurring in other regions (Harrison & Booth, 2007; Mangubhai & Harrison, 2008). This might explain why in Los Roques and other coral reefs (Apprill & Rappé, 2011), phytoplankton blooms were not detected after massive coral spawning events.

In addition to the intensity and magnitude of spawning events across regions, other factors might influence the variability in the phytoplankton response. In this study, the sampling days within spawning periods (Before, During and After) were established based on expected spawning dates for Acropora palmata, as this species marks the beginning of the multi-specific coral spawning event in Caribbean shallow waters. However, variations in spawning intensity and time, as reported for this coral species (unpublished data by MW Miller & AM Szmant in Key Largo, Florida and La Parguera, Puerto Rico in 2000, Japp between the 1970s and 1980s and Miller in 2001), are likely to affect the observed response of microphytoplankton assemblages between years and among spawning periods. In particular, Acropora palmata larval abundance suggested a late start in spawning for most reefs in 2008 (i.e., +5 days instead of +2–3 days, Table S3). The abundance of A. palmata larvae in the plankton at the time of this study (Zubillaga, 2010) strongly suggested that coral spawning at Los Roques was larger in 2008 than in 2007, but in 2008 it lagged with respect to our sampling periods in most of the reefs (Table S3). This would have resulted in a During period being similar to a Before period, both with little spawn material; explaining the differences in composition and abundance of microphytoplankton observed between 2007 and 2008 and the observed increase in abundance of diatoms and chlorophyll a concentration at one of the SW reefs only in 2008. Days and reefs with higher abundance of coral larvae (Zubillaga, 2010) coincided with diatoms being more abundant, a group that requires higher inorganic nutrient concentrations compared to cyanophytes and dinoflagellates (Reynolds, 2006). In contrast, the higher abundance of heterotrophic dinoflagellates in 2007 at various reefs, and of mixotrophic dinoflagellates at the GR reef site in 2008, might indicate a subtle shift in these trophic groups, favouring heterotrophy.

A clear pattern of the role of microheterotrophy in the microphytoplankton response to larvae abundance, as it has been reported for phytoplankton biomass and primary productivity (Eyre, Glud & Patten, 2008; Glud, Eyre & Patten, 2008), was apparently hidden by the high variability observed among reefs. Previous work in which biomass, primary productivity and/or respiration in sediment and water have peaked shortly after the release of spawned material (Eyre, Glud & Patten, 2008; Glud, Eyre & Patten, 2008) were carried out only at one site, on a reef flat. Thus, an examination of these processes (i.e., phytoplankton primary production, biomass and species composition in relation to nutrient re-mineralization) at larger spatial scales in reefs might yield large variability as it was found in this study.

In addition to the amount of material spawned, other local (reef) scale-dependent processes such as re-mineralization of organic matter, might be related to the variations in chla and microphytoplankton observed among reefs in nine days. The response of photosynthetic microalgae to the input of labile organic compounds is mediated by the metabolic activity of the microbial reef community, often called the microbial loop (Anderson & Ducklow, 2001; Pomeroy et al., 2007; Nelson et al., 2013). This process entails great variability as it may operate on time scales of hours to days (Carlson et al., 2002) and it depends on many factors, from benthic composition to oceanographic characteristics of the reefs (Nelson et al., 2011). Re-mineralization of coral spawn materials is mostly carried out in reef sediments (Westneat & Resing, 1988; Haas et al., 2011; Haas et al., 2013), whereas the role played by bacterioplankton in the process vary from immediate and significant (Wild et al., 2008) to lagged and marginal (Apprill & Rappé, 2011). Thus, the temporal response of the microphytoplankton community to an organic matter input might be controlled by how fast the re-mineralized material is incorporated into the water column through benthic-pelagic coupling (Eyre, Glud & Patten, 2008; Patten, Harrison & Mitchell, 2008; Nelson et al., 2011; Apprill & Rappé, 2011). In this sense, during coral spawning events in the Gulf of Mexico, Horne (2012) observed only small changes of nutrient concentrations in water, which suggests a low re-mineralization activity in the water column, but he also reported an increased abundance of dinoflagellates of the genus Ceratium spp. Thus, in our study, reef-specific differences in the re-mineralization activity could have contributed towards the variability found at the reef scale in the structure of diatoms assemblages.

In this study, functional group and species-specific patterns of microphytoplankton related to the spawning period were observed at some reefs. During and After the coral spawning in 2007, the increased abundance in Protoperidinium heterotrophic species, and the concomitant decrease of the mixotrophic Neoceratium lineatum, may constitute a response to the coral spawning via abundance of prey. The abundance of heterotrophic dinoflagellates found in this study was very similar to that found by Horne (2012) in the Gulf of Mexico 2 days after the coral spawning. However, at some reefs in Los Roques, the abundance of mixotrophic dinoflagellates was almost double that reported by Horne (2012). Some heterotrophic dinoflagellate species prey preferentially upon heterotrophic bacteria and protozoans and the eggs and larvae of metazoans (Jeong, 1994; Jeong et al., 2007; Jeong et al., 2010), whereas mixotrophic species more frequently prey upon small-size microalgae like haptophytes, crysophytes, picophytes, raphidophytes (not assessed on this study), chlorophytes, autotrophic dinoflagellates, some diatom species and autotrophic bacteria (Li, Stoecker & Coats, 2000; Jeong et al., 2005a; Jeong et al., 2005b; Jeong et al., 2005c; Berge, Hansen & Moestrup, 2008). The negative correlation between the abundance of heterotrophic dinoflagellates and one of their grazed species, the diatom Rhizosolenia imbricata (Willén, 1991), might indicate that preferential feeding occurred in detriment of autotrophic microalgae species. Consistent with this, heterotrophic and mixotrophic dinoflagellates are recognized to have important effects on plankton abundance and composition as microheterotrophic grazers (Lessard & Swift, 1985). Thus, despite increased Chla concentrations, grazing activity by heterotrophic dinoflagellates might explain the lack of a significant increase in diatom abundance in relation to spawning periods at some reefs in 2007.

In addition to grazing, microheterotrophic phytoplankton (i.e., mixo- and heterotrophic dinoflagellates among other protozoans) maintain nutrient demand through re-mineralization of organic matter in oceanic waters year-round, and their contribution to dissolved inorganic nutrients might be higher than that of zooplankton (Bode et al., 2004). Oligotrophic coral reefs have an efficient recycling of nutrients (Szmant, 2002; Wild et al., 2008) and are able to incorporate a massive coral-spawning organic matter input in less than 6 days through the benthic microbial loop. Because sediment re-mineralized nitrogen is not readily returned to the water column (Eyre, Glud & Patten, 2008), microheterotrophic re-mineralization within the microphytoplankton community might be mediating the response of autotrophic groups to the input of that organic matter. Although the role of these grazers in the re-mineralization of spawned materials in coral reefs has not been evaluated, the positive correlation between the diatom Cerataulina pelagica—whose abundance has been related to increase on dissolved nitrate and ammonium availability in the water column (Härnström, Karunasagar & Godhe, 2009)—and the heterotrophic dinoflagellates of the genus Protoperidinium, might suggest an important role of the latter on the availability of dissolved inorganic nutrients at the SW reefs. A similar increase in dinoflagellate abundance coinciding with a decrease in that of the diatoms was found 2 days after coral spawning in the Gulf of Mexico (Horne, 2012).

At Los Roques, high abundance of small rather than medium and large size diatoms, especially in the relatively pristine SW reefs, might indicate an effective nutrient cycling as cell size in microalgae has an influence on nutrient preference and uptake (Koike, Holm-Hansen & Biggs, 1986; Stolte et al., 1994), with smaller phytoplankters favoured over larger ones in systems driven by regenerated nutrients and rapid cycling of organic matter (Caroppo, 2000). Similarly, the peak abundance in diatoms coinciding with that of mixotrophic dinoflagellates in 2008 at GR might also respond to re-mineralization of spawned materials by the latter. However, the consistency of this pattern was not captured in our 2008 sampling probably due to lagged spawn in the other reefs. Also, sampling days for each time period (i.e., Before-During-After) were relatively short (3 days each), and these spawning periods were contiguous. If the response of the larger size phototrophs of the phytoplankton to spawned organic matter is mediated by microheterotrophy, as suggested by our results, more days between sampling periods could have been better for detecting changes in the microphytoplankton assemblages.

Observed microphytoplankton density, although well within previous reports for Dos Mosquises in Los Roques and other oligotrophic waters (e.g., González, 1989; Spiniello, 1996; Madera & Furderer, 1997), revealed spatiotemporal differences of the microphytoplankton community in Los Roques at scales previously unexplored. Differences in plankton assemblages known to occur between the NE and SW sectors of the archipelago during the dry season from November to June (González, 1989; Casanova, Zoppy de Roa & Montiel, 2007) also hold true for August and September during this study, a period with a weak hydrodynamic influence in the area (Casanova, Zoppy de Roa & Montiel, 2007). Our results also indicated a strong reef-based variability within previously thought homogenous sectors (Casanova, Zoppy de Roa & Montiel, 2007) and a daily variability within otherwise considered constant seasons (González, 1989; Spiniello, 1996; Madera & Furderer, 1997). The presence of the cyanophyte Anabaena sp. in August and September on the NE reefs in 2007 constitutes the first report of this species in Los Roques, indicating episodic freshwater inputs, consistent with human settlement at Gran Roque. Algae from this genus can produce toxins of public health concern (De Figueiredo et al., 2004); however, their disappearance in samples from consecutive days suggests unfavourable water conditions for these populations and/or short water residence time. Toxin-producing dinoflagellate species were also identified, but in low abundance.

While our study lacked of direct biomass measures through time, studies elsewhere have not targeted compositional changes on microalgae assemblages. Therefore, a full picture of changes in phytoplankton across regions and reefs in relation to coral spawning is still lacking. Our results highlighted the importance of including taxonomical and functional changes in larger size phytoplankton fractions when evaluating nutrient fluxes and primary and secondary pelagic productivity in coral reefs in relation to coral spawning events, as well as the importance of replication at reef spatial scales for generalizing observed patterns. Manipulative experiments to establish the effect of microheterotrophic grazing and nutrient re-mineralization activity by dinoflagellates on the response of the phytoplankton assemblages to organic matter inputs constitute a logical next step to better comprehend the role of dinoflagellates in reef waters during a nutrient input. Concomitantly, specific processes related to inter-annual variability need to be assessed through the sampling of non-spawning times as well in order to better discriminate phytoplankton responses to coral spawning events. Similarly, large-scale operating oceanographic and/or climatic processes not yet assessed on the study site may also have an important role as structuring factors, as suggested by the amount of inter-annual variability observed in the microphytoplankton structure and composition and the daily patterns of SST and Chla concentrations.

This study evaluated the structure of larger size fractions of the phytoplankton during coral spawning for the first time in the Caribbean, showing variations in dinoflagellate assemblages and its trophic groups, as well as in some diatom species. These variations highlighted the potential impact of heterotrophic dinoflagellates on coral reefs primary productivity and organic matter cycling. Thus, by examining the structure of microphytoplankton at multiple spatiotemporal scales in Los Roques archipelago, this study provides a first approximation of the effect of coral spawning on these assemblages.

Supplemental Information

Table S1 Microalgae species identified in water samples from four reefs, in nine consecutive days before, during and after coral spawning in 2007 and 2008 at Los Roques, Venezuela, Southern Caribbean

Click here for additional data file.

Table S2 Simper species total density

Click here for additional data file.

Data S1 Average and total density by taxonomic group and SIMPER species

Click here for additional data file.

Table S3 Acropora palmata larvae abundance values

Acropora palmata larvae abundance, measured through specific antibody signal during spawning times evaluated in 2007 and 2008 at Los Roques, Venezuela, Southern Caribbean. Values shown represent the average absorbance of the four replicates taken at each reef.

Click here for additional data file.

The authors would like to thank Alejandra Hernández, Adriana Humanes, Sebastián Rodríguez, Denise Debrot, María Isabel Reyes, Helios Martínez, Melanie Van Solt, Julie Pérez, Aldo Cróquer, Federico Pisani, Julia Dávila and the staff of Fundación Científica Los Roques for the help during the samplings, and also, Professor Juan José Cruz for the help provided with equipment.

Additional Information and Declarations

Competing Interests

Author Contributions

Data Availability

The authors declare there are no competing interests.

Francoise Cavada-Blanco conceived and designed the experiments, performed the experiments, analyzed the data, wrote the paper, prepared figures and/or tables, reviewed drafts of the paper.

Ainhoa L. Zubillaga and Carolina Bastidas conceived and designed the experiments, performed the experiments, contributed reagents/materials/analysis tools, wrote the paper, reviewed drafts of the paper.

The following information was supplied regarding data availability:

The raw data was supplied as Data S1.

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
