# Peer review of "Microphytoplankton variations during coral spawning at Los Roques, Southern Caribbean"

_PeerJ, doi:10.7717/peerj.1747_

## Round 0.1 · original submission · Major Revisions

Both reviewers acknowledge the value of this work and that this study adds significantly to our knowledge, however they are also highlighting the need for additional specific environmental measurements as identified in your discussion. Both have also mentioned the manuscript is at times lengthy and I would encourage you to be more concise where achievable.

Reviewer 1 ·

Basic reporting

Line 46 (introduction), please clarify/provide more detail about 'local stressors'
Line 75 (introduction), please provide approximate size range for 'small-size fractions'
Line 93 (introduction), I think that by 'surplus of nutrients' you mean that coral spawning events causes input of nutrients, please clarify what you mean by 'represent'
Line 163 (sample collection), by 'basis of Acroporids', do you mean 'basis that Acoporids'
Line 170 (sample collection), please provide further explanation of 'specific antibody signals'
Line 343 (discussion), not sure about 'OK?' please check this.
Line 510 (discussion), please provide reference for '...a period with low hydrodynamic characteristics in the area'. I think that you mean 'weak influences', or similar, please clarify.
Caption for Figure 1, please include mention of Venezuela, Carribean
Caption for Figure 2, please make reference to graph scales being different for each plot.
Caption for Figure 3, please add a space between 'group' and 'size'. Also, please make reference to graph scales being different for each plot.
Caption for Figure 4, order of 'average richness' and 'shannon diversity' in caption does not match order in graphs.

Experimental design

The authors examine phytoplankton assembleges through two coral spawning events over a 2 year period, at four reef sites for the first time in the Los Roques archipelago Caribbean

I think that more data over a longer time period could provide more insight, also coupled with specific environmental measurements but the authors have identified these issues/information gaps in their discussion.

There may be other information available to put the phytoplankton results into context that the authors may have not considered. (eg. ocean currents/circulation patterns in the area, water depth at sampling sites and weather conditions in the lead up to and during sample collection. Were any concurrent measurements of temperature and salinty made during sampling? Where available, these factors should be included in the article.

Validity of the findings

the authors provide a good baseline for phytoplankton succession and temporal and spatial variations at different reefs in this archipelago. The discussion is lengthy, however, it provides a review of the knowledge gaps within this study and the field in general.

Can the authors speculate as to why, in general, higher phytoplankton densities were found in the NE sector? is this linked to any human input sources?

Reviewer 2 ·

Basic reporting

No comments

Experimental design

No comments

Validity of the findings

No comments

Additional comments

The manuscript titled “Microphytoplankton variations during spawning at Lost Rogues, southern Caribbean” presents an important study examining the temporal and spatial variations in microphytoplankton assemblages over an expected coral spawning in the Caribbean. The authors clearly outline the hierarchical (nested) experimental design of sampling, that is, 4 replicate water samples over three times - before (3 days), during (3 days) and after (3days) expected spawning, and at 4 reefs over 2 consecutiveyears. Major findings included i) high interannual variability in microphytoplankton abundance and composition ii) high spatial variability amongst reefs; iii) an increase in heterotrophic and mixotrophic dinoflagellates related to a decrease in C. pelagica and an increase in R. imbricate during and after spawning at some reefs.

Whilst the study adds substantially to our limited knowledge on the temporal and spatial changes in microphytoplankton assemblages in relation to nutrient additions in tropical reefs, I believe the manuscript is limited in its presentation significance due to the following reasons:

1. There is no concurrent nutrient data, nor water quality data with which the observed microphytoplankton data could be assessed against.
2. There is no data presented on the coral spawning which may assist in explaining the weak, or delayed, microphytoplankton response.
3. Phytoplankton response to nutrient inputs may take longer than three days after an event, and although this is acknowledged by the authors, it was not considered in the experimental design.
4. In general the Introduction and Discussion could be more concise and therefore reduced.

Some specific comments are as follows:
• Line 1 The first sentence is confusing and perhaps could be divided into two points.
• Line 49 Koop et al 2001 should come before Heil et al 2004.
• Line 140 locations in text and Fig. 1 should be indicated by sexagesimals or decimal units.
• Lines 175 to Lines 181 should be placed in a table for ease of reading.
• How were microphytoplankton measured/size fractionated?
• What statistical package/version was used for analyses?
• Be careful throughout when using “species” when in fact it is probably more accurate to use the word “taxa” or “genera” – check throughout manuscript.
• Figure 2 should be divided into A - F and legend and text should reflect this. This should also be done for Figures 3 – 7.
• Line 238 Most of the diatoms identified (Figure 3A) are more accurately described as being in the range of 6 - 10 m and 16 - 20 m as this is why the size fractionation was done; similarly most of the dinoflagellates identified are in the range of 16 - 20 m (Figure 3B)
• Table 1 Should highlight significant P levels with “*”
• Line 298 I think should read “P. minutum” not “P. minimum”
• Line 382 “OK?” I am not sure this belongs here??
• Suplementary Table 1. “sp.” should not be italised. Many genera need “spp.” instead of “sp.” unless you are dealing with one species belonging to this genus eg. “Chaetoceros sp.” should read “Chaetoceros spp.”
• Incidently, I am not sure that any measure of biodiversity (eg. Margalef’s species richness index or Shannon diversity index) is an appropriate measure for microphytoplankton as the diversity of a genus such as Chaetoceros remains unknown in this case. That is, how can you measure diversity when there are many taxa identified as “Chaetoceros spp.”
• Pseudo-nitzschia delicatissima and P. seriata cannot be identified using light microscopy. Revise to read “P. delicatissima group” and “P. seriata” group.
• “Noctiluca scintillans”
• Supplementary Table 2 – units?

---

## Round 0.2 · accepted · Accept

Publication will be subject to the additional final minor changes requested by reviewer 1.

Reviewer 1 ·

Basic reporting

Abstract
Line 26 – sentence beginning ‘However…” is too long. Please rewrite.
Line 26 – please clarify chlorophyll. I belive that this would also be surface values. please confirm and this should be clarified throughout text.
Line 27 – please clarify ‘better matched’

Introduction
Line 85 – do you mean larger species of microaglae? Please clarify

Results
Please check reporting of ‘r2’, should this be ‘r2’?
Text refers to figure 4a and b but capital letters are used in the figures
Line 347 – how was the variability between years measured? Please clarify. Looking at figure 8, what appears to be mean and SD lines are not described in the caption. This is alsy the case for figure 9. There seems to be some confusion in the text Figure 8 is CHL but is referred to as SST in the text. This is also the case for figure 9 (CHL), it is referred to as SST in the text.
Line 354 - Please clarify the values used in table 4. Is this the average over the 9 days? This should also be clearer in the text.

Discussion
Line 376 – do you mean ‘within a few days’?
Line 373-379 this sentence beginning ‘A lack of..” is too long. Please rewrite.
Line 440 – should this be ‘spawned material’?
Line 523 – it reads like there may be a word missing? Suggest ‘…..well within previously reported levels…’

Figures
Please check the abbreviations for august in the figures.
Please check the x-axis headings for ‘E’ in the CHL plots (figure 9). This looks like something has been distorted. The plots would be clearer if the panel ‘E’ dates aligned with the panels above.

Experimental design

no further comments

Validity of the findings

no further comments

Additional comments

Thanks to the authors for including the original recommendations to their manuscript. There are some minor items that need further clarification (see above).